# Mini-batch kernel $k$-means

**Anonymous**

## Abstract

We present the first mini-batch kernel $k$-means algorithm. Our algorithm achieves an order of magnitude improvement in running time compared to the full batch algorithm, with only a minor negative effect on the quality of the solution. Specifically, a single iteration of our algorithm requires only $O(n(k + b))$ time, compared to $O(n^2)$ for the full batch kernel $k$-means, where $n$ is the size of the dataset and $b$ is the batch size.

We provide a theoretical analysis for our algorithm with an early stopping condition and show that if the batch is of size $\Omega((\gamma/\epsilon)^2 \log(n\gamma/\epsilon))$, the algorithm must terminate within $O(\gamma^2/\epsilon)$ iterations with high probability, where $\gamma$ is the bound on the norm of points in the dataset in feature space, and $\epsilon$ is a threshold parameter for termination. Our results hold for any reasonable initialization of centers. When the algorithm is initialized with the $k$-means++ initialization scheme, it achieves an approximation ratio of $O(\log k)$.

Many popular kernels are *normalized* (e.g., Gaussian, Laplacian), which implies $\gamma = 1$. For these kernels, taking $\epsilon$ to be a constant and $b = \Theta(\log n)$, our algorithm terminates within $O(1)$ iterations where each iteration takes time $O(n(\log n + k))$.

## 1  Introduction

Mini-batch methods are among the most successful tools for handling huge datasets for machine learning. Notable examples include Stochastic Gradient Descent (SGD) and mini-batch $k$-means [27]. Mini-batch $k$-means [27] is one of the most popular clustering algorithms used in practice [21].

While $k$-means is widely used due to it's simplicity and fast running time, it requires the data to be *linearly separable* to achieve meaningful clustering. Unfortunately, many real-world datasets do not have this property. One way to overcome this problem is to project the data into a high, even *infinite*, dimensional space (where it is hopefully linearly separable) and run $k$-means on the projected data using the "kernel-trick".

Kernel $k$-means achieves significantly better clustering compared to $k$-means in practice. However, its running time is considerably slower. Surprisingly, prior to our work there was no attempt to speed up kernel $k$-means using a mini-batch approach.

**Problem statement**    We are given an input (dataset), $X = \{x_i\}_{i=1}^{n}$, of size $n$ and a parameter $k$ representing the number of clusters. A kernel for $X$ is a function $K : X \times X \to \mathbb{R}$ that can be realized by inner products. That is, there exists a Hilbert space $\mathcal{H}$ and a map $\phi : X \to \mathcal{H}$ such that $\forall x, y \in X, \langle \phi(x), \phi(y) \rangle = K(x, y)$. We call $\mathcal{H}$ the *feature space* and $\phi$ the *feature map*.

In kernel $k$-means the input is a dataset $X$ and a kernel function $K$ as above. Our goal is to find a set $\mathcal{C}$ of $k$ centers (elements in $\mathcal{H}$) such that the following goal function is minimized:

$$\frac{1}{n} \sum_{x \in X} \min_{c \in \mathcal{C}} \|c - \phi(x)\|^2.$$

38th Conference on Neural Information Processing Systems (NeurIPS 2024).

Equivalently we may ask for a partition of $X$ into $k$ parts, keeping $\mathcal{C}$ implicit.[1]

**Lloyd's algorithm**   The most popular algorithm for (non kernel) $k$-means is Lloyd's algorithm, often referred to as the $k$-means algorithm [17]. It works by randomly initializing a set of $k$ centers and performing the following two steps: (1) Assign every point in $X$ to the center closest to it. (2) Update every center to be the mean of the points assigned to it. The algorithm terminates when no point is reassigned to a new center. This algorithm is extremely fast in practice but has a worst-case exponential running time [3, 30].

**Mini-batch $k$-means**   To update the centers, Lloyd's algorithm must go over the entire input at every iteration. This can be computationally expensive when the input data is extremely large. To tackle this, the mini-batch $k$-means method was introduced by [27]. It is similar to Lloyd's algorithm except that steps (1) and (2) are performed on a batch of $b$ elements sampled uniformly at random with repetitions, and in step (2) the centers are updated slightly differently. Specifically, every center is updated to be the weighted average of its current value and the mean of the points (in the batch) assigned to it. The parameter by which we weigh these values is called the *learning rate*, and its value differs between centers and iterations. The larger the learning rate, the more a center will drift towards the new batch cluster mean.

**Lloyd's algorithm in feature space**   Implementing Lloyd's algorithm in feature space is challenging as we cannot explicitly keep the set of centers $\mathcal{C}$. Luckily, we can use the kernel function together with the fact that centers are always set to be the mean of cluster points to compute the distance from any point $x \in X$ in feature space to any center $c = \frac{1}{|A|} \sum_{y \in A} \phi(y)$ as follows:

$$\|\phi(x) - c\|^2 = \langle \phi(x) - c, \phi(x) - c \rangle = \langle \phi(x), \phi(x) \rangle - 2\langle \phi(x), c \rangle + \langle c, c \rangle$$

$$= \langle \phi(x), \phi(x) \rangle - 2\langle \phi(x), \frac{1}{|A|} \sum_{y \in A} \phi(y) \rangle + \langle \frac{1}{|A|} \sum_{y \in A} \phi(y), \frac{1}{|A|} \sum_{y \in A} \phi(y) \rangle,$$

where $A$ can be any subset of the input $X$. While the above can be computed using only kernel evaluations, it makes the update step significantly more costly than standard $k$-means. Specifically, the complexity of the above may be quadratic in $n$ [9].

**Mini-batch kernel $k$-means**   Applying the mini-batch approach for kernel $k$-means is even more difficult because the assumption that cluster centers are always the mean of some subset of $X$ in feature space no longer holds.

In Section 4 we present our algorithm and derive a recursive expression that allows us to compute the distances of all points to current cluster centers (in feature space). Our algorithm implements this by updating a data structure that maintains the inner products between the data and centers in feature space. This means the running time of each iteration of our algorithm is only $O(n(b + k))$ compared to $O(n^2)$ for the full-batch algorithm.

In Section 5 we go on to provide theoretical guarantees for our algorithm. This is somewhat tricky for mini-batch algorithms due to their stochastic nature, as they may not even converge to a local-minima. To overcome this hurdle, we take the approach of [26] and answer the question: how long does it take mini-batch kernel $k$-means to terminate with an *early stopping condition*. Specifically, we terminate the algorithm when the improvement on the batch drops below some user provided parameter, $\epsilon$. Early stopping conditions are very common in practice (e.g., sklearn[21]).

Applying the $k$-means++ initialization scheme for our initial centers implies we achieve the same approximation ratio, $O(\log k)$ in expectation, as the full-batch algorithm. The approximation guarantee of $k$-means++ is guaranteed already in the initialization phase (Theorem 3.1 in [4]), and the execution of Lloyd's algorithm following initialization can only improve the solution. We show that w.h.p[2] the global goal function is decreasing throughout our execution which implies that the approximation guarantee remains the same.

---

[1]A common variant of the above is when every $x \in X$ is assigned a weight $w_x \in \mathbb{R}^+$ and we aim to minimize $\sum_{x \in X} w_x \cdot \min_{c \in \mathcal{C}} \|c - \phi(x)\|^2$. Everything that follows, including our results, can be easily generalized to the weighted case. We present the unweighted case to improve readability.

[2]This is usually taken to be $1 - 1/n^p$ for some constant $p \geq 1$. For our case, it holds that $p = 1$, however, this can be amplified arbitrarily by increasing the batch size by a multiplicative constant factor.

While our general approach is similar to [26], we must deal with the fact that $\mathcal{H}$ may have an *infinite* dimension. The guarantees of [26] depend on the dimension of the space in which $k$-means is executed, which is unacceptable in our case. We overcome this by parameterizing our results by a new parameter $\gamma = \max_{x \in X} \|\phi(x)\|$. We note that for normalized kernels, such as the popular Gaussian and Laplacian kernels, it holds that $\gamma = 1$. We show that if the batch size is $\Omega((\gamma/\epsilon)^2 \log(n\gamma/\epsilon))$ then w.h.p. our algorithm terminates in $O(\gamma^2/\epsilon)$ iterations. Our theoretical results are summarised in Theorem 1 (where Algorithm 1 is presented in Section 4).

**Theorem 1.** *The following holds for Algorithm 1:*

1. *Each iteration takes $O(n(b + k))$ time,*

2. *If $b = \Omega((\gamma/\epsilon)^2 \log(n\gamma/\epsilon))$ then it terminates in $O(\gamma^2/\epsilon)$ iterations with high probability,*

3. *When initialized with k-means++ it achieve a $O(\log k)$ approximation ratio in expectation.*

Our result improves upon [26] significantly when a normalized kernel is used since Theorem 1 doesn't depend on the input dimension. Our algorithm copes better with non linearly separable data and requires a smaller batch size ($\widetilde{O}(1/\epsilon^2)$ vs $\widetilde{O}((d/\epsilon)^2)))$[3] for normalized kernels. This is particularly apparent with high dimensional datasets such as MNIST[16] where the dimension squared is already nearly ten times the number of datapoints.

The learning rate we use, suggested in [26], differs from the standard learning rate of sklearn in that it does not go to 0 over time. Unfortunately, this new learning rate is non-standard and [26] did not present experiments comparing their learning rate to that of sklearn.

In Section 6 we evaluate our results experimentally both with the learning rate of [26] and that of sklearn. We also fill the experimental gap left in [26] by evaluating (non-kernel) mini-batch $k$-means with their new learning rate compared to that of sklearn. To allow a fair empirical comparison, we run each algorithm for a fixed number of iterations without stopping conditions. Our results are as follows:

- Mini-batch kernel $k$-means is significantly faster than full-batch kernel $k$-means, while achieving solutions of similar quality, which are superior to the non-kernel version.

- The learning rate of [26] results in solutions with better quality both for mini-batch kernel $k$-means and mini-batch $k$-means.

## 2   Related work

Until recently, mini-batch $k$-means was only considered with a learning rate going to 0 over time. This was true both in theory [29, 27] and practice [21]. Recently, [26] proposed a new learning which does not go to 0 over time, and showed that if the batch is of size $\tilde{\Omega}((d/\epsilon)^2)$, mini-batch $k$-means must terminate within $O(d/\epsilon)$ iterations with high probability, where $d$ is the dimension of the input, and $\epsilon$ is a threshold parameter for termination.

A popular approach to deal with the slow running time of kernel $k$-means is constructing a *coreset* of the data. A coreset for kernel $k$-means is a weighted subset of $X$ with the guarantee that the solution quality on the coreset is close to that on the entire dataset up to a $(1 + \epsilon)$ multiplicative factor. There has been a long line of work on coresets for $k$-means an kernel k-means [25, 10, 5], and the current state-of-the-art for kernel k-means is due to [13]. They present a coreset algorithm with a nearly linear (in $n$ and $k$) construction time which outputs a coreset of size $poly(k\epsilon^{-1})$.

In [7] the authors only compute the kernel matrix for uniformly sampled set of $m$ points from $X$. Then they optimize a variant of kernel $k$-means where the centers are constrained to be linear combinations of the sampled points. The authors do no provide worst case guarantees for the running time or approximation of their algorithm.

Another approach to speed up kernel $k$-means is by computing an approximation for the kernel matrix. This can be done by computing a low dimensional approximation for $\phi$ (without computing $\phi$ explicitly)[23, 8, 6], or by computing a low rank approximation for the kernel matrix [19, 31].

---

[3]Where $\widetilde{O}$ hides factors that are logarithmic in $d, n, 1/\epsilon$.

Kernel sparsification techniques construct sparse approximations of the full kernel matrix in sub-quadratic time. For smooth kernel functions such as the polynomial kernel, [22] presents an algorithm for constructing a $(1 + \epsilon)$-spectral sparsifier for the full kernel matrix with a nearly linear number of non-zero entries in nearly linear time. For the gaussian kernel, [18] show how to construct a weaker, cluster preserving sparsifier using a nearly linear number of kernel density estimation querries.

We note that our results are *complementary* to coresets, dimensionality reduction, and kernel sparsification, in the sense that we can compose our method with these techniques.

## 3   Preliminaries

Throughout this paper we work with ordered tuples rather than sets, denoted as $Y = (y_i)_{i \in [\ell]}$, where $[\ell] = \{1, \ldots, \ell\}$. To reference the $i$-th element we either write $y_i$ or $Y[i]$. It will be useful to use set notations for tuples such as $x \in Y \iff \exists i \in [\ell], x = y_i$ and $Y \subseteq Z \iff \forall i \in [\ell], y_i \in Z$. When summing we often write $\sum_{x \in Y} g(x)$ which is equivalent to $\sum_{i=1}^{\ell} g(Y[i])$.

We borrow the following notation from [14] and generalize it to Hilbert spaces. For every $x, y \in \mathcal{H}$ let $\Delta(x, y) = \|x - y\|^2$. We slightly abuse notation and and also write $\Delta(x, y) = \|\phi(x) - \phi(y)\|^2$ when $x, y \in X$ and $\Delta(x, y) = \|\phi(x) - y\|^2$ when $x \in X, y \in \mathcal{H}$ (similarly when $x \in \mathcal{H}, y \in X$). For every finite tuple $S \subseteq X$ and a vector $x \in \mathcal{H}$ let $\Delta(S, x) = \sum_{y \in S} \Delta(y, x)$. Let us denote $\gamma = \max_{x \in X} \|\phi(x)\|$. Let us define for any finite tuple $S \subseteq X$ the center of mass of the tuple as $cm(S) = \frac{1}{|S|} \sum_{x \in S} \phi(x)$.

We now state the kernel $k$-means problem using the above notation.

**Kernel $k$-means**   We are given an input $X = (x_i)_{i=1}^n$ and a parameter $k$. Our goal is to (implicitly) find a tuple $\mathcal{C} \subseteq \mathcal{H}$ of $k$ centers such that the following goal function is minimized: $\frac{1}{n} \sum_{x \in X} \min_{C \in \mathcal{C}} \Delta(x, C)$.

Let us define for every $x \in X$ the function $f_x : \mathcal{H}^k \to \mathbb{R}$ where $f_x(\mathcal{C}) = \min_{C \in \mathcal{C}} \Delta(x, C)$. We can treat $\mathcal{H}^k$ as the set of $k$-tuples of vectors in $\mathcal{H}$. We also define the following function for every tuple $A = (a_i)_{i=1}^\ell \subseteq X$: $f_A(\mathcal{C}) = \frac{1}{\ell} \sum_{i=1}^\ell f_{a_i}(\mathcal{C})$. Note that $f_X$ is our original goal function.

We extensive use of the notion of *convex combination*:

**Definition 2.** *We say that $y \in \mathcal{H}$ is a* convex combination *of $X$ if $y = \sum_{x \in X} p_x \phi(x)$, such that $\forall x \in X, p_x \geq 0$ and $\sum_{x \in X} p_x = 1$.*

## 4   Our Algorithm

We present our pseudo-code as Algorithm 1. It requires an initial set of cluster centers such that every center is a convex combination of $X$. This guarantees that all subsequent centers are also a convex combination of $X$. Note that if we initialize the centers using the kernel version of $k$-means++, this is indeed the case.

Algorithm 1 proceeds by repeatedly sampling a batch of size $b$ (the batch size is a parameter). For the $i$-th batch the algorithm (implicitly) updates the centers using the learning rate $\alpha_j^i$ for center $j$. Note that the learning rate may take on different values for different centers, and may change between iterations. Finally, the algorithm terminates when the progress on the batch is below $\epsilon$, a user provided parameter. While our termination guarantees (Section 5) require a specific learning rate, it does not affect the running time of a single iteration, and we leave it as a parameter for now.

**Recursive distance update rule**   While for (non kernel) $k$-means the center updates and assignment of points to clusters is straightforward, this is tricky for kernel $k$-means and even harder for mini-batch kernel $k$-means. Specifically, how do we overcome the challenge that we do not maintain the centers explicitly?

To assign points to centers in the $(i + 1)$-th iteration, it is sufficient to know $\|\phi(x) - \mathcal{C}_{i+1}^j\|^2$ for every $j$. If we can keep track of this quantity through the execution of the algorithm, we are done.

---

**Algorithm 1:** Mini-batch kernel $k$-means with early stopping

---

1 **Input:**
  - Dataset $X = (x_i)_{i=1}^n$, batch size $b$, early stopping parameter $\epsilon$
  - Initial centers $(\mathcal{C}_1^j)_{j=1}^k$ where $\mathcal{C}_1^j$ is a convex combination of $X$ for all $j \in [k]$

2 **for** $i = 1$ *to* $\infty$ **do**
3      Sample $b$ elements, $B_i = (y_1, \ldots, y_b)$, uniformly at random from $X$ (with repetitions)
4      **for** $j = 1$ *to* $k$ **do**
5         $B_i^j = \left\{ x \in B_i \mid \arg\min_{\ell \in [k]} \Delta(x, \mathcal{C}_i^\ell) = j \right\}$
6         $\alpha_i^j$ is the learning rate for the $j$-th cluster for iteration $i$
7         $\mathcal{C}_{i+1}^j = (1 - \alpha_i^j)\mathcal{C}_i^j + \alpha_i^j cm(B_i^j)$
8      **if** $f_{B_i}(\mathcal{C}_{i+1}) - f_{B_i}(\mathcal{C}_i) < \epsilon$ **then** Return $\mathcal{C}_{i+1}$

---

Let us derive a *recursive* expression for the distances

$$\|\phi(x) - \mathcal{C}_{i+1}^j\|^2 = \langle \phi(x), \phi(x) \rangle - 2\langle \phi(x), \mathcal{C}_{i+1}^j \rangle + \langle \mathcal{C}_{i+1}^j, \mathcal{C}_{i+1}^j \rangle.$$

We first expand $\langle \phi(x), \mathcal{C}_{i+1}^j \rangle$,

$$\langle \phi(x), \mathcal{C}_{i+1}^j \rangle = \langle \phi(x), (1 - \alpha_i^j)\mathcal{C}_i^j + \alpha_i^j cm(B_i^j) \rangle = (1 - \alpha_i^j)\langle \phi(x), \mathcal{C}_i^j \rangle + \alpha_i^j \langle \phi(x), cm(B_i^j) \rangle.$$

Then we expand $\langle \mathcal{C}_{i+1}^j, \mathcal{C}_{i+1}^j \rangle$,

$$\langle \mathcal{C}_{i+1}^j, \mathcal{C}_{i+1}^j \rangle = \langle (1 - \alpha_i^j)\mathcal{C}_i^j + \alpha_i^j cm(B_i^j), (1 - \alpha_i^j)\mathcal{C}_i^j + \alpha_i^j cm(B_i^j) \rangle$$
$$= (1 - \alpha_i^j)^2 \langle \mathcal{C}_i^j, \mathcal{C}_i^j \rangle + 2\alpha_i^j(1 - \alpha_i^j)\langle \mathcal{C}_i^j, cm(B_i^j) \rangle + (\alpha_i^j)^2 \langle cm(B_i^j), cm(B_i^j) \rangle.$$

Assuming that $\langle \mathcal{C}_i^j, \mathcal{C}_i^j \rangle$ and $\langle \phi(x), \mathcal{C}_i^j \rangle$ are known for all $j \in [k]$ and for all $x \in X$, we can compute $\langle \mathcal{C}_{i+1}^j, \mathcal{C}_{i+1}^j \rangle$ and $\langle \phi(x), \mathcal{C}_{i+1}^j \rangle$ for all $j \in [k]$ and $x \in X$, which implies we can compute the distances from any point in the batch to all centers.

We now bound the running time of a single iteration of the outer loop in Algorithm 1. Let us denote $b_i^j = \left| B_i^j \right|$ and recall that $cm(B_i^j) = \frac{1}{b_i^j} \sum_{y \in B_i^j} \phi(y)$. Therefore, computing $\langle \phi(x), cm(B_i^j) \rangle = \frac{1}{b_i^j} \sum_{y \in B_i^j} \langle \phi(x), \phi(y) \rangle$ requires $O(b_i^j)$ time. Similarly, computing $\langle cm(B_i^j), cm(B_i^j) \rangle$ requires $O((b_i^j)^2)$ time. Let us now bound the time it requires to compute $\langle \phi(x), \mathcal{C}_{i+1}^j \rangle$ and $\langle \mathcal{C}_{i+1}^j, \mathcal{C}_{i+1}^j \rangle$.

Assuming we know $\langle \phi(x), \mathcal{C}_i^j \rangle$ and $\langle \mathcal{C}_i^j, \mathcal{C}_i^j \rangle$, updating $\langle \phi(x), \mathcal{C}_{i+1}^j \rangle$ for all $x \in X, j \in [k]$ requires $O(n(b+k))$ time. Specifically, the $\langle \phi(x), \mathcal{C}_i^j \rangle$ term is already known from the previous iteration and we need to compute $\alpha_i^j \langle \phi(x), cm(B_i^j) \rangle$ for every $x \in X, j \in [k]$ which requires $n \sum_{j \in [k]} b_i^j = nb$ time. Finally, updating $\langle \phi(x), \mathcal{C}_{i+1}^j \rangle$ for all $x \in X, j \in [k]$ requires $O(nk)$ time.

Updating $\langle \mathcal{C}_{i+1}^j, \mathcal{C}_{i+1}^j \rangle$ requires $O(b^2 + kb)$ time. Specifically, $\langle \mathcal{C}_i^j, \mathcal{C}_i^j \rangle$ is known from the previous iteration and computing $\langle cm(B_i^j), cm(B_i^j) \rangle$ for all $j \in [k]$ requires $O(\sum_{j \in [k]}(b_i^j)^2) = O(b^2)$ time. Computing $\langle \mathcal{C}_i^j, cm(B_i^j) \rangle$ for all $j \in [k]$ requires time $O(b)$ using $\langle \phi(x), \mathcal{C}_i^j \rangle$ from the previous iteration. Therefore, the total running time of the update step (assigning points to new centers) is $O(n(b+k))$. To perform the update at the $(i+1)$-th step we only need $\langle \phi(x), \mathcal{C}_i^j \rangle, \langle \mathcal{C}_i^j, \mathcal{C}_i^j \rangle$, which results in a space complexity of $O(nk)$. This completes the first claim of Theorem 1.

## 5 Termination guarantee

**Section preliminaries**    We introduce the following definitions and lemmas to aid our proof of the second claim of Theorem 1.

**Lemma 3.** *For every $y$ which is a convex combination of $X$ it holds that $\|y\| \leq \gamma$.*

*Proof.* The proof follows by a simple application of the triangle inequality:

$$\|y\| = \|\sum_{x \in X} p_x \phi(x)\| \leq \sum_{x \in X} \|p_x \phi(x)\| = \sum_{x \in X} p_x \|\phi(x)\| \leq \sum_{x \in X} p_x \gamma = \gamma.$$

$\square$

**Lemma 4.** *For any tuple of $k$ centers $\mathcal{C} \subset \mathcal{H}^d$ which are a convex combination of points in $X$, it holds that $\forall A \subseteq X, f_A(\mathcal{C}) \leq 4\gamma^2$.*

*Proof.* It is sufficient to upper bound $f_x$. Combining that fact that every $C \in \mathcal{C}$ is a convex combination of $X$ with the triangle inequality, we have that

$$\forall x \in X, f_x(\mathcal{C}) \leq \max_{C \in \mathcal{C}} \Delta(x, C) = \Delta(x, \sum_{y \in X} p_y \phi(y))$$

$$= \|\phi(x) - \sum_{y \in X} p_y \phi(y)\|^2 \leq (\|\phi(x)\| + \|\sum_{y \in X} p_y \phi(y)\|)^2 \leq 4\gamma^2. \qquad \square$$

We state the following simplified version of an Azuma bound for Hilbert space valued martingales from [20], followed by a standard Hoeffding bound.

**Theorem 5** ([20]). *Let $\mathcal{H}$ be a Hilbert space and let $Y_0, ..., Y_m$ be a $\mathcal{H}$-valued martingale, such that $\forall 1 \leq i \leq m, \|Y_i - Y_{i-1}\| \leq a_i$. It holds that $Pr[\|Y_m - Y_0\| \geq \delta] \leq e^{\Theta\left(\frac{\delta^2}{\sum_{i=1}^m a_i^2}\right)}$.*

**Theorem 6** ([12]). *Let $Y_1, ..., Y_m$ be independent random variables such that $\forall 1 \leq i \leq m, E[Y_i] = \mu$ and $Y_i \in [a_{min}, a_{max}]$. Then*

$$Pr\left(\left|\frac{1}{m} \sum_{i=1}^m Y_k - \mu\right| \geq \delta\right) \leq 2e^{-2m\delta^2/(a_{max}-a_{min})^2}.$$

The following lemma provide concentration guarantees when sampling a *batch*.

**Lemma 7.** *Let $B$ be a tuple of $b$ elements chosen uniformly at random from $X$ with repetitions. For any fixed tuple of $k$ centers, $\mathcal{C} \subseteq \mathcal{H}$ which are a convex combination of $X$, it holds that: $Pr[|f_B(\mathcal{C}) - f_X(\mathcal{C})| \geq \delta] \leq 2e^{-b\delta^2/2\gamma^2}$.*

*Proof.* Let us write $B = (y_1, \ldots, y_b)$, where $y_i$ is a random element selected uniformly at random from $X$ with repetitions. For every such $y_i$ define the random variable $Z_i = f_{y_i}(\mathcal{C})$. These new random variables are IID for any fixed $\mathcal{C}$. It also holds that $\forall i \in [b], E[Z_i] = \frac{1}{n} \sum_{x \in X} f_x(\mathcal{C}) = f_X(\mathcal{C})$ and that $f_B(\mathcal{C}) = \frac{1}{b} \sum_{x \in B} f_x(\mathcal{C}) = \frac{1}{b} \sum_{i=1}^b Z_i$.

Applying the Hoeffding bound (Theorem 6) with parameters $m = b, \mu = f_X(\mathcal{C}), a_{max} - a_{min} \leq 4\gamma^2$ (due to Lemma 4) we get that: $Pr[|f_B(\mathcal{C}) - f_X(\mathcal{C})| \geq \delta] \leq 2e^{-b\delta^2/2\gamma^2}$. $\square$

For any tuple $S \subseteq X$ and some tuple of cluster centers $\mathcal{C} = (\mathcal{C}^\ell)_{\ell \in [k]} \subset \mathcal{H}$, $\mathcal{C}$ implies a *partition* $(S^\ell)_{\ell \in [k]}$ of the points in $S$. Specifically, every $S^\ell$ contains the points in $S$ closest to $\mathcal{C}^\ell$ (in $\mathcal{H}$) and every point in $S$ belongs to a single $\mathcal{C}^\ell$ (ties are broken arbitrarily). We state the following useful observation:

**Observation 8.** *Fix some $A \subseteq X$. Let $\mathcal{C}$ be a tuple of $k$ centers, $S = (S^\ell)_{\ell \in [k]}$ be the partition of $A$ induced by $\mathcal{C}$ and $\overline{S} = (\overline{S}^\ell)_{\ell \in [k]}$ be any other partition of $A$. It holds that $\sum_{j=1}^k \Delta(S^j, \mathcal{C}^j) \leq \sum_{j=1}^k \Delta(\overline{S}^j, \mathcal{C}^j)$.*

Recall that $\mathcal{C}_i^j$ is the $j$-th center in the beginning of the $i$-th iteration of Algorithm 1 and $(B_i^\ell)_{\ell \in [k]}$ is the partition of $B_i$ induced by $\mathcal{C}_i$. Let $(X_i^\ell)_{\ell \in [k]}$ be the partition of $X$ induced by $\mathcal{C}_i$.

We now have the tools to analyze Algorithm 1 with the learning rate of [26]. Specifically, we assume that the algorithm executes for at least $t$ iterations, the learning rate is $\alpha_i^j = \sqrt{b_i^j/b}$, where $b_i^j = \left|B_i^j\right|$, and the batch size is $b = \Omega((\gamma/\epsilon)^2 \log(nt))$. We show that the algorithm must terminate within $t = O(\gamma/\epsilon)$ steps w.h.p. Plugging $t$ back into $b$, we get that a batch size of $b = \Omega((\gamma/\epsilon)^2 \log(n\gamma/\epsilon))$ is sufficient.

**Proof outline** We note that when sampling a batch it holds w.h.p that $f_{B_i}(\mathcal{C}_i)$ is close to $f_{X_i}(\mathcal{C}_i)$ (Lemma 7). This is due to the fact that $B_i$ is sampled after $\mathcal{C}_i$ is fixed. If we could show that $f_{B_i}(\mathcal{C}_{i+1})$ is close $f_{X_i}(\mathcal{C}_{i+1})$ then combined with the fact that we make progress of at least $\epsilon$ on the batch we can conclude that we make progress of at least some constant fraction of $\epsilon$ on the entire dataset.

Unfortunately, as $C_{i+1}$ depends on $B_i$, getting the above guarantee is tricky. To overcome this issue we define the auxiliary value $\overline{\mathcal{C}}_{i+1}^j = (1 - \alpha_i^j)\mathcal{C}_i^j + \alpha_i^j cm(X_i^j)$. This is the $j$-th center at step $i + 1$ if we were to use the entire dataset for the update, rather than just a batch. Note that this is only used in the analysis and not in the algorithm. Note that $\overline{\mathcal{C}}_{i+1}$ only depends on $\mathcal{C}_i$ and $X$ and is independent of $B_i$ (i.e., we can fix its value before sampling $B_i$). As $\overline{\mathcal{C}}_{i+1}$ does not depend on $B_i$ we use $\overline{\mathcal{C}}_{i+1}$ instead of $\mathcal{C}_{i+1}$ in the above analysis outline. We show that for our choice of learning rate it holds that $\overline{\mathcal{C}}_{i+1}, \mathcal{C}_{i+1}$ are sufficiently close, which implies that $f_X(\mathcal{C}_{i+1}), f_X(\overline{\mathcal{C}}_{i+1})$ and $f_{B_i}(\mathcal{C}_{i+1}), f_{B_i}(\overline{\mathcal{C}}_{i+1})$ are also sufficiently close. That is, $\overline{\mathcal{C}}_{i+1}$ acts as a proxy for $\mathcal{C}_{i+1}$. Combining everything together we get our desired result.

We start with the following useful observation, which will allow us to use Lemma 3 to bound the norm of the centers by $\gamma$ throughout the execution of the algorithm.

**Observation 9.** *If $\forall j \in [k], \mathcal{C}_1^j$ is a convex combination of $X$ then $\forall i > 1, j \in [k], \mathcal{C}_i^j, \overline{\mathcal{C}}_i^j$ are also a convex combinations of $X$.*

Let us state the following useful lemma from [14]. Although their proof is for Euclidean spaces, it goes through for Hilbert spaces. We provide the proof in the appendix for completeness.

**Lemma 10** ([14])**.** *For any set $S \subseteq X$ and any $C \in \mathcal{H}$ it holds that $\Delta(S, C) = \Delta(S, cm(S)) + |S|\,\Delta(C, cm(S))$.*

We use the above to prove the following useful lemma.

**Lemma 11.** *For any $S \subseteq X$ and $C, C' \in \mathcal{H}$ which are convex combinations of $X$, it holds that: $|\Delta(S, C') - \Delta(S, C)| \leq 2\gamma\,|S|\,\|C - C'\|$.*

*Proof.* Using Lemma 10 we get that $\Delta(S, C) = \Delta(S, cm(S)) + |S|\,\Delta(cm(S), C)$ and that $\Delta(S, C') = \Delta(S, cm(S)) + |S|\,\Delta(cm(S), C')$. Thus, it holds that $|\Delta(S, C') - \Delta(S, C)| = |S| \cdot |\Delta(cm(S), C') - \Delta(cm(S), C)|$. Let us write

$$
\begin{aligned}
&|\Delta(cm(S), C') - \Delta(cm(S), C)| \\
&= |\langle cm(S) - C', cm(S) - C'\rangle - \langle cm(S) - C, cm(S) - C\rangle| \\
&= |-2\langle cm(S), C'\rangle + \langle C', C'\rangle + 2\langle cm(S), C\rangle - \langle C, C\rangle| \\
&= |2\langle cm(S), C - C'\rangle + \langle C' - C, C' + C\rangle| \\
&= |\langle C - C', 2cm(S) - (C' + C)\rangle| \\
&\leq \|C - C'\|\|2cm(S) - (C' + C)\| \leq 4\gamma\|C - C'\|.
\end{aligned}
$$

Where in the last transition we used the Cauchy-Schwartz inequality, the triangle inequality, and the fact that $C, C', cm(S)$ are convex combinations of $X$ and therefore their norm is bounded by $\gamma$. $\square$

Now we show that due to our choice of learning rate, $\mathcal{C}_{i+1}^j$ and $\overline{\mathcal{C}}_{i+1}^j$ are sufficiently close.

**Lemma 12.** *It holds w.h.p that $\forall i \in [t], j \in [k], \|\mathcal{C}_{i+1}^j - \overline{\mathcal{C}}_{i+1}^j\| \leq \frac{\epsilon}{20\gamma}$.*

*Proof.* Note that $\mathcal{C}_{i+1}^j - \overline{\mathcal{C}}_{i+1}^j = \alpha_i^j(cm(B_i^j) - cm(X_i^j))$. Let us fix some iteration $i$ and center $j$. To simplify notation, let us denote: $X' = X_i^j, B' = B_i^j, b' = b_i^j, \alpha' = \alpha_i^j$. Although $b'$ is a random variable, in what follows we treat it as a fixed value (essentially conditioning on its value). As what follows holds for *all* values of $b'$ it also holds without conditioning due to the law of total probabilities.

For the rest of the proof, we assume $b' > 0$ (if $b' = 0$ the claim holds trivially). Let us denote by $\{Y_\ell\}_{\ell=1}^{b'}$ the sampled points in $B'$. Note that a randomly sampled element from $X$ is in $B'$ if and only if it is in $X'$. As batch elements are sampled uniformly at random with repetitions from $X$,

conditioning on the fact that an element is in $B'$ means that it is distributed uniformly over $X'$. Note that $\forall \ell, E[\phi(Y_\ell)] = \frac{1}{|X'|} \sum_{x \in X'} \phi(x) = cm(X')$ and $E[cm(B')] = \frac{1}{b'} \sum_{\ell=1}^{b'} E[\phi(Y_\ell)] = cm(X')$.

Let us define the following martingale: $Z_r = \sum_{\ell=1}^{r} (\phi(Y_\ell) - E[\phi(Y_\ell)])$. Note that $Z_0 = 0$, and when $r > 0$, $Z_r = \sum_{\ell=1}^{r} \phi(Y_\ell) - r \cdot cm(X')$. It is easy to see that this is a martingale:

$$E[Z_r \mid Z_{r-1}] = E[\sum_{\ell=1}^{r} \phi(Y_\ell) - r \cdot cm(X') \mid Z_{r-1}] = Z_{r-1} + E[\phi(Y_r) - cm(X') \mid Z_{r-1}] = Z_{r-1}.$$

Let us now bound the differences

$$\|Z_r - Z_{r-1}\| = \|\phi(Y_r) - cm(X')\| \leq \|\phi(Y_r)\| + \|cm(X')\| \leq 2\gamma.$$

Now we may use Azuma's inequality: $Pr[\|Z_{b'} - Z_0\| \geq \delta] \leq e^{\Theta(\frac{\delta^2}{\gamma^2 b'})}$. Let us now divide both sides of the inequality by $b'$ and set $\delta = \frac{b'\epsilon}{20\gamma\alpha'}$. We get

$$Pr[\|cm(B') - cm(X')\| \geq \frac{\epsilon}{20\gamma\alpha'}] = Pr[\|\frac{1}{b'} \sum_{\ell=1}^{b'} \phi(Y_\ell) - cm(X')\| \geq \frac{\epsilon}{20\gamma\alpha'}] \leq e^{\Theta(\frac{b'\epsilon^2}{(\gamma\alpha')^2})}.$$

Using the fact that $\alpha' = \sqrt{b'/b}$ together with the fact that $b = \Omega((\gamma/\epsilon)^2 \log(nt))$ (for an appropriate constant) we get that the above is $O(1/ntk)$. Finally, taking a union bound over all $t$ iterations and all $k$ centers per iteration completes the proof. $\qquad\square$

We can now bound the goal function when cluster centers are close using Lemma 12.

**Lemma 13.** *Fix some $A \subseteq X$. It holds w.h.p that $\forall i \in [t], |f_A(\overline{C}_{i+1}) - f_A(C_{i+1})| \leq \epsilon/5$.*

*Proof.* Let $S = (S^\ell)_{\ell \in [k]}, \overline{S} = (\overline{S}^\ell)_{\ell \in [k]}$ be the partitions induced by $C_{i+1}, \overline{C}_{i+1}$ on $A$. Let us expand the expression

$$f_A(\overline{C}_{i+1}) - f_A(C_{i+1}) = \frac{1}{|A|} \sum_{j=1}^{k} \Delta(\overline{S}^j, \overline{C}_{i+1}^j) - \Delta(S^j, C_{i+1}^j)$$

$$\leq \frac{1}{|A|} \sum_{j=1}^{k} \Delta(S^j, \overline{C}_{i+1}^j) - \Delta(S^j, C_{i+1}^j)$$

$$\leq \frac{1}{|A|} \sum_{j=1}^{k} 4\gamma |S^j| \|\overline{C}_{i+1}^j - C_{i+1}^j\| \leq \frac{1}{|A|} \sum_{j=1}^{k} |S^j| \epsilon/5 = \epsilon/5.$$

Where the first inequality is due to Observation 8, the second is due Lemma 11 and finally we use Lemma 12 together with the fact that $\sum_{j=1}^{k} |S^j| = |A|$. Using the same argument we also get that $f_A(C_{i+1}) - f_A(\overline{C}_{i+1}) \leq \epsilon/5$, which completes the proof. $\qquad\square$

Let us state the following useful lemma.

**Lemma 14.** *It holds w.h.p that for every $i \in [t]$,*

$$f_X(\overline{C}_{i+1}) - f_X(C_{i+1}) \geq -\epsilon/5, \tag{1}$$

$$f_{B_i}(C_{i+1}) - f_{B_i}(\overline{C}_{i+1}) \geq -\epsilon/5, \tag{2}$$

$$f_X(C_i) - f_{B_i}(C_i) \geq -\epsilon/5, \tag{3}$$

$$f_{B_i}(\overline{C}_{i+1}) - f_X(\overline{C}_{i+1}) \geq -\epsilon/5. \tag{4}$$

*Proof.* The first two inequalities follow from Lemma 13. The last two are due to Lemma 7 by setting $\delta = \epsilon/5, B = B_i$:

$$Pr[|f_{B_i}(C) - f_X(C)| \geq \delta] \leq 2e^{-b\delta^2/2\gamma^2} = e^{-\Theta(b\epsilon^2/\gamma^2)} = e^{-\Omega(\log(nt))} = O(1/nt).$$

Where the last inequality is due to the fact that $b = \Omega((\gamma/\epsilon)^2 \log(nt))$ (for an appropriate constant). The above holds for either $C = C_i$ or $C = \overline{C}_{i+1}$. Taking a union bound over all $t$ iterations we get the desired result. $\qquad\square$

**Putting everything together**  We wish to lower bound $f_X(\mathcal{C}_i) - f_X(\mathcal{C}_{i+1})$. We write the following, where the $\pm$ notation means we add and subtract a term:

$$
\begin{aligned}
f_X(\mathcal{C}_i) - f_X(\mathcal{C}_{i+1}) &= f_X(\mathcal{C}_i) \pm f_{B_i}(\mathcal{C}_i) - f_X(\mathcal{C}_{i+1}) \\
&\geq f_{B_i}(\mathcal{C}_i) - f_X(\mathcal{C}_{i+1}) - \epsilon/5 = f_{B_i}(\mathcal{C}_i) \pm f_{B_i}(\mathcal{C}_{i+1}) - f_X(\mathcal{C}_{i+1}) - \epsilon/5 \\
&\geq f_{B_i}(\mathcal{C}_{i+1}) - f_X(\mathcal{C}_{i+1}) + 4\epsilon/5 \\
&= f_{B_i}(\mathcal{C}_{i+1}) \pm f_{B_i}(\overline{\mathcal{C}}_{i+1}) \pm f_X(\overline{\mathcal{C}}_{i+1}) - f_X(\mathcal{C}_{i+1}) + 4\epsilon/5 \geq \epsilon/5.
\end{aligned}
$$

Where the first inequality is due to inequality (3) in Lemma 14 ($f_X(\mathcal{C}_i) - f_{B_i}(\mathcal{C}_i) \geq -\epsilon/5$), the second is due to the stopping condition of the algorithm ($f_{B_i}(\mathcal{C}_i) - f_{B_i}(\mathcal{C}_{i+1}) > \epsilon$), and the last is due to the remaining inequalities in Lemma 14. The above holds w.h.p over all of the iterations of the algorithms. We conclude that when $b = \Omega((\gamma/\epsilon)^2 \log(\gamma n/\epsilon))$, w.h.p. the algorithm terminates within $t = O(\gamma^2/\epsilon)$ iterations. This complete the proof of the second claim of Theorem 1.

## 6   Experiments

We evaluate our mini-batch algorithm on the following datasets:

- **MNIST:** The MNIST dataset [16] has 70,000 grayscale images of handwritten digits (0 to 9), each image being 28x28 pixels. When flattened, this gives 784 features.
- **PenDigits:** The PenDigits dataset [1] has 10992 instances, each represented by an 16-dimensional vector derived from 2D pen movements. The dataset has 10 labelled clusters, one for each digit.
- **Letters:** The Letters dataset [28] has 20,000 instances of letters from 'A' to 'Z', each represented by 16 features. The dataset has 26 labelled clusters, one for each letter.
- **HAR:** The HAR dataset [2] has 10,299 instances collected from smartphone sensors, capturing human activities like walking, sitting, and standing. Each instance is described by 561 features. The dataset has 6 labelled clusters, corresponding to different types of physical activities.

We compare the following algorithms: (full batch) kernel k-means, mini-batch kernel k-means with the learning rate of [26], mini-batch kernel k-means with the learning rate of sklearn, mini-batch (non-kernel) k-means with the learning rate of [26], mini-batch (non-kernel) k-means with the learning rate of sklearn. We evaluate our results with batch sizes: 1024, 256, 64, and 16. We execute every algorithm for 200 iterations. For the kernel variants we apply the Gaussian kernel: $K(x,y) = e^{-\|x-y\|^2/\kappa}$, where the $\kappa$ parameter is set using the heuristic of [31] followed by some manual tuning (exact values appear in the supplementary materials). We repeat every experiment 10 times and present the average Adjusted Rand Index (ARI) [11, 24] and Normalized Mutual Information (NMI) [15] scores for every dataset. All experiments were conducted on a MacBook Pro equipped with an M2 Max chip and 96 GB of RAM. We present partial results in Figure 1 and the full results in the appendix. Error bars in the plot measure the standard deviation.

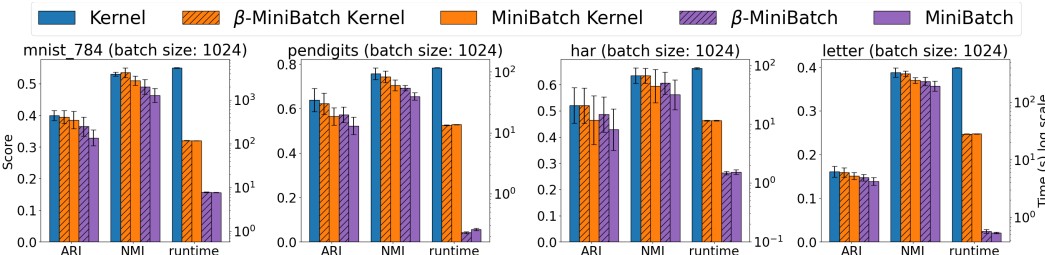

Figure 1: Our results for a batch size of size 1024. We use the $\beta$ prefix to denote the algorithm uses the learning rate of [26].

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

# A    Omitted proofs

**Proof of Lemma 10**

*Proof.*

$$
\begin{aligned}
\Delta(S, C) &= \sum_{x \in S} \Delta(x, C) = \sum_{x \in S} \langle x - C, x - C \rangle \\
&= \sum_{x \in S} \langle (x - cm(S)) + (cm(S) - C), (x - cm(S)) + (cm(S) - C) \rangle \\
&= \sum_{x \in S} \Delta(x, cm(S)) + \Delta(C, cm(S)) + 2\langle x - cm(S), cm(S) - C \rangle \\
&= \Delta(S, cm(S)) + |S|\, \Delta(C, cm(S)) + \sum_{x \in S} 2\langle x - cm(S), cm(S) - C \rangle \\
&= \Delta(S, cm(S)) + |S|\, \Delta(C, cm(S)),
\end{aligned}
$$

where the last step is due to the fact that

$$
\begin{aligned}
\sum_{x \in S} \langle x - cm(S), cm(S) - C \rangle &= \langle \sum_{x \in S} x - |S|\, cm(S), cm(S) - C \rangle \\
&= \langle \sum_{x \in S} x - \frac{|S|}{|S|} \sum_{x \in S} x, cm(S) - C \rangle = 0.
\end{aligned}
$$

$\square$

# B    Full experimental results

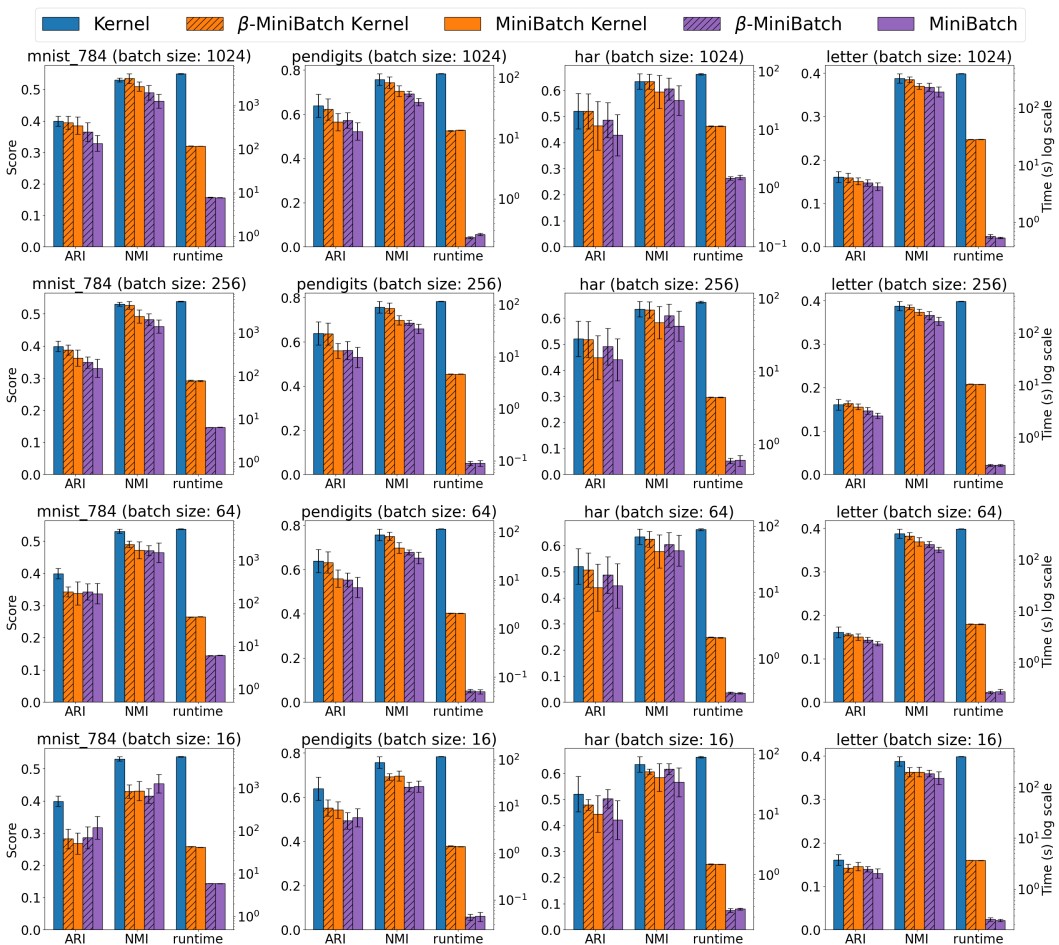

Figure 2: Our results for all batch sizes. We use the $\beta$ prefix to denote the algorithm uses the learning rate of [26].

