# OpenReview forum: "Mini-batch kernel $k$-means"
_NeurIPS.cc/2024/Conference — Submitted to NeurIPS 2024_

### Official Review · Reviewer_mEhQ · 2024-07-05

**Soundness:** 3
**Presentation:** 3
**Contribution:** 2
**Rating:** 6
**Confidence:** 4

**Summary:**

The authors present the first mini-batch algorithm for kernel k-means. The algorithm itself is simple and works the way one would expect mini-batch kernel k-means to work. The authors improve the running time of an iteration of kernel k-means from $O(n^2)$ to $O(n(k+b))$ for the mini-batch version of the algorithm. Additionally, they show that using a specific learning rate function, there is an upper bound on the number of iterations of the algorithm.

The main challenge in the design of this algorithm is to keep track of the intermediate centers as storing the points in feature space is infeasible, as they are updated iteratively as in Lloyd's algorithm. For this, the authors design a recursive update rule to keep track of the quantity $\| \phi(x) - C_i^j \|^2$ for each iteration $i$ and each center $j$. They show that in a new iteration this quantity can be updated by considering the distance of each point in the dataset to the centers of mass of the clusters in the mini-batch and the previous centers.

Finally, the authors provide an experimental study of the mini-batch algorithm on four datasets and compare it to a non-kernel mini-batch algorithm and the full kernel k-means algorithm.

**Strengths:**

- The algorithm offers improved running time bounds that are interesting to practitioners using kernel k-means in practice. It is also the first algorithm for mini-batch kernel k-means.
- The main theorem's bound on the number of iterations is nice to have and a good follow up to paper [26].
- The theoretical analysis is cleanly written and easy to follow.

**Weaknesses:**

- The techniques, while elegant are not particularly novel in terms of theory. The proofs mostly follow from analyzing the inner product terms in the k-means formulation.
- A number of the proofs in the main body of the paper could have been moved to the appendix, as they do not give the reader more of an understand of the big picture and are very detail specific.
- There is no discussion of the experimental results.
- While the authors state the paper is mostly theoretical, I believe this algorithm is mostly interesting to practicioners and therefore a more thorough focus on the experimental evaluation with more parameters, additional datasets and thorough discussion would strengthen the paper in my eyes.

**Questions:**

Throughout the paper you assume that the computation of the kernel function $\langle \phi(x), \phi(y)\rangle$ can be done in constant time. Is there a particular reason for this? It appears to me as though many kernel functions would take time $\Theta(d)$ to compute.

In the plot for the experimental results, it is hard for me to understand the influence of the batch size on the running time, can you elaborate why we don't see bigger changes in many of the plots, and why there is such a large jump in the har dataset going from 256 to 1024?

**Limitations:**

The authors included a checklist in the appendix of the paper, but have not discussed practical limitations in the main body of the paper.

---

> ### Author Rebuttal · Authors · 2024-08-05
>
> Thank you for your questions and suggested improvements.
>
> We agree with your observation that our main contribution is for practitioners, and that our work can benefit from additional experiments. We ran additional experiments on graph kernels (see our main rebuttal for more details). Please let us know if there are any additional experimental results you would like to see.
>
> On your question about the time to evaluate kernel functions, yes you are correct. For kernels such as the Gaussian kernel it takes time $O(d)$. However, often the kernel matrix is the input to the problem itself.  As such, papers often assume oracle access to the kernel matrix. We will add a note about this with the runtime if you have to start from scratch.
>
>
> To your question about the lack of difference in runtime across batch sizes, we decided to include the cost of constructing the full kernel matrix in the runtime to be fair to the full kernel k-means algorithm. Had we not included this, the difference in runtime would have been even more stark. In our new experiments we present the kernel construction times separately so the difference is visible. In practice , there are other techniques that can be used to alleviate this up front cost when starting from scratch such as composing our algorithm with a corset or kernel sparsification algorithm.

---

> > ### Comment · Reviewer_mEhQ · 2024-08-12
> >
> > Thank you for your clarification and running the additional experiments.

---

### Official Review · Reviewer_gVj8 · 2024-07-10

**Soundness:** 3
**Presentation:** 2
**Contribution:** 2
**Rating:** 5
**Confidence:** 4

**Summary:**

This paper proposes the first mini-batch kernel $k$-means algorithm, which significantly reduces running time compared to the previous kernel $k$-means methods relying on the full datasets. With the proposed mini-batch kernel $k$-means algorithm, each iteration can be executed in time $O(n(k+b))$, improving the complexity of $O(n^2)$ for fully-batch methods. The authors also provide theoretical guarantees, ensuring that the algorithm can terminate (reach a convergence) within $O(\gamma^2/\epsilon)$ iterations with high probability, where $\gamma$ is the bound on the norm of points in the feature space. When initialized with the $k$-means++ seeding method, the algorithm achieves an $O(logk)$-approximation. Experimental  evaluations confirm that the mini-batch kernel k-means algorithm performs significantly faster than its full-batch counterpart while maintaining solution quality.

**Strengths:**

The proposed algorithm achieves significant improvements on time complexity compared with full-batch kernel $k$-means methods.

The paper provides theoretical analysis, ensuring the algorithm's termination and performance bounds if initialized with the $k$-means++ seeding method.

**Weaknesses:**

The techniques used in this paper are largely based on the work of [1]. Mini-batch $k$-means method is not new for clustering problem. The main contribution of this paper is to combine the idea of mini-batch $k$-means with the kernel $k$-means versions. It should be noted that the theoretical bounds given in this paper are not entirely novel.

There are some technical issues in the proofs (details see questions),  potentially undermining the theoretical guarantees.


[1] Gregory Schwartzman. Mini-batch $k$-means terminates within $O(d/\epsilon)$ iterations. ICLR 2024.

**Questions:**

**Question 1:** For Mini-batch $k$-means methods, the centers are only updated based on a small sample of the whole datasets without calculating the exact clustering cost of the whole dataset. Thus, why the authors claim that "the execution of Lloyd's algorithm following initialization can only improve the solution and the why the approximation guarantee can remain the same for $k$-means++ method?

**Question 2:** In line 8 of Algorithm 1, I think the condition of returning the center set should be "if $f_{B_i}(C_{i}) - f_{B_i}(C_{i+1})) < \epsilon$, then return $C_{i+1}$", since $f_{B_i}(C_{i})$ is intuitivelly larger than than $f_{B_i}(C_{i+1})) $.

**Question 3:** In the proof of Lemma 7, since $a_{max} - a_{min} \le 4\gamma^2$, the probability bound by using Hoeffding Inequality should be $P_r[|f_B(C) - f_X(C)| \ge \delta] \le 2e^{-b\delta^2 / 8\gamma^4}$ instead of $2e^{-b\delta^2/2\gamma^2}$. If I am wrong, please point out.

**Question 4:** In Theorem 5, I think there is an negative sign in the probability bound, i.e., $e^{\Theta(\delta^2 / \sum_{i=1}^{m}a_i^2)}$ should be $e^{\Theta(-\delta^2 / \sum_{i=1}^{m}a_i^2)}$.

**Question 5:** In the proof of Lemma 12, I think the sample size is not enough for obtaining a success probability of around $\Omega(1 - 1/ntk)$.  $b$ should be at least $b = \Omega((\gamma^2/\epsilon)^2log(nt))$.

**Question 6:** In experiments, how to choose the kernel functions to achieve the desired performance?

**Question 7:** The technical proofs of termination bounds given in this paper are quite similar to that for Mini-batch $k$-means proposed in ICLR 2024 (see [1] above). It seems that the only differences between these two work is the initial bound for the clustering cost difference, one is $d$ for ICLR 2024 and one is the maximum norm distance for this work. Can the authors give detailed discussions on the differences between these two work and the main technical contribution of the proposed kernel $k$-means method?


Minor Comments:

In page 6, "The following lemma provide concentration guarantees when sampling a batch" should be "The following lemma provides concentration guarantees when sampling a batch".

**Limitations:**

Although this paper mainly gives theoretical results for clustering problems, it lacks discussions on broader impact as required by the NeurIPS guidelines.

---

> ### Author Rebuttal · Authors · 2024-08-05
>
> Thank you for taking the time to comb through are paper. You have been very generous with your time and have given us some new ideas to think about.
>
> We understand your concern regarding the novelty of the theoretical analysis. However, as reviewer mEhQ pointed out, our main contribution is in making a big step towards making kernel k-means usable in practice by reducing the running time by an order of magnitude.
> Furthermore, our theoretical analysis shows that both the number of iterations and the batch size of mini-batch kernel k-means are better than the (non-kernel) mini-batch k-means for normalized kernels, which we found to be quite surprising.
>
> To respond to your questions:
>
> - Q1: Since Kmeans++ gives an O(log(k)) approximation ratio in expectation and whp our algorithm either makes progress or stops, we are guaranteed to match the same approximation in expectation.
> - Q2: Yes, we have $f_{B_i}(C_{i+1})$ and $f_{B_i}(C_{i})$ the wrong way around.
> - Q3/ Q5: Yes, you are correct. Thanks for spotting this! This means we pick up the extra factor of $\gamma^2$ in the batch size as you point out in Q5. This does not affect our results for normalized kernels.
> This extra term is interesting in it's own right: a higher order dependence on $\gamma$ actually helps shrink batch sizes for kernel functions induced by graphs (see our main rebuttal for more details) as they often have $\gamma<<1$ and are often more useful in practice compared to the gaussian/laplacian kernel functions with $\gamma=1$. See our new results for a comparison.
> - Q4: Yes that is a typo thanks.
> - Q5: Answered above.
> - Q6: Choosing a good kernel function is still a bit of a (dark) art. In our submission we followed a heuristic from the literature [Mahoney and Wang, ref 31 in the paper] to pick a reasonable bandwidth parameter for the gaussian kernel function. However, we found that using the kernel function induced by a knn-graph is often a lot easier as its parameter, the number of neighbours, is far easier to tune. See our results for a comparison.
> - Q7: The main technical difference between our theoretical results and those of Schwartzman is indeed in the generalization to Hilbert spaces, the introduction of the parameter gamma, and using Hilbert space valued martingales. We agree that the general proof follows the same outline and that our main contribution is the conceptual idea of using mini-batch methods to speed up kernel k-means.
>
> Thanks for spotting the typo on page 6. Regarding impact, please suggest what discussions on broader impact you would like to see.

---

### Official Review · Reviewer_hmC1 · 2024-07-13

**Soundness:** 3
**Presentation:** 3
**Contribution:** 2
**Rating:** 5
**Confidence:** 3

**Summary:**

In this paper, the authors propose the first mini-batch kernel k-means clustering algorithm. It is a variant of Lloyd's algorithm that was introduced by Sculley that takes a batch of random b points instead of the full set of points and a weighted avaerage with the current centers while updating the centers. This paper attempts to translate this idea in the *kernel* k-means setting. The resulting algorithm has the same approximation guarantee as the original k-means but it terminates faster and consumes less time per iteration.

Their analysis follows the recipe of Scwartzman who used an early stopping condition when the improvement on the batch drops below some user-provided parameter. The main challenge in the kernel setting is that the underlying Hilbert space could be large or even infinite-dimensional. This is prohibitive as Scwartzman's bound on the number of iterations depends on the dimension. The authors bypass this by instead giving a bound on the Hilbert norm of the points which can be bounded in practice for example using normalized kernels.

**Strengths:**

The authors coduct detailed experiments that compares their algorithm favorably with the prior works. Specfically, the ARI and NMI scores were noticably better across a variety of 4 datasets.

I liked the paper. I think it has a decent theoretical and experimental contribution.

**Weaknesses:**

New ideas are limited. Mostly an adaptation of Scwartzman's work in the kernel setting

**Questions:**

None.

**Limitations:**

None.

---

> ### Author Rebuttal · Authors · 2024-08-05
>
> Thank you for taking the time to go through our paper.
>
> We understand your concern regarding the novelty of the theoretical analysis. However, as reviewer mEhQ pointed out, our main contribution is in making a big step towards making kernel k-means usable in practice by reducing the running time by an order of magnitude.
> Furthermore, our theoretical analysis shows that both the number of iterations and the batch size of mini-batch kernel k-means are better than the (non-kernel) mini-batch k-means for normalized kernels, which we found to be quite surprising. We think minibatch methods will be vital for practical kernel k-means. Do you have any suggestions for practical improvements?

---

> > ### Comment · Reviewer_hmC1 · 2024-08-14
> > **Read the rebuttal**
> >
> > I have read the rebuttal from the authors. I don't have any suggestions for practical improvements. I'll maintain my score.

---

### Official Review · Reviewer_AkgB · 2024-07-13

**Soundness:** 2
**Presentation:** 3
**Contribution:** 2
**Rating:** 4
**Confidence:** 2

**Summary:**

The article presents the first mini-batch kernel k-means algorithm, which significantly improves running time compared to the full batch kernel $k$-means with only a minor negative effect on solution quality. The proposed algorithm runs in $O(n(k+b))$ time per iteration, as opposed to $O(n^2)$ for the full-batch version. The authors provide theoretical guarantees for the algorithm's performance, demonstrating that it terminates within $O(\gamma^2/\epsilon)$ iterations with high probability when the batch size is $\Omega((\gamma/\epsilon)^2 \log(n\gamma/\epsilon))$. Experimental results confirm the efficiency and effectiveness of the algorithm.

**Strengths:**

Improved Efficiency: The mini-batch approach drastically reduces the running time from $O(n^2)$ to $O(n(k+b))$ per iteration, making it feasible to handle large datasets.

Theoretical Guarantees: The algorithm includes a thorough theoretical analysis, ensuring termination within a specific number of iterations and providing an approximation ratio when using $k$-means++ initialization.

Flexibility with Kernels: The algorithm works well with popular normalized kernels (e.g., Gaussian, Laplacian), making it versatile for various applications.

Practical Relevance: Early stopping conditions align with practical machine learning workflows, increasing the algorithm's usability in real-world scenarios.

**Weaknesses:**

Approximation Quality: While the solution quality is comparable to the full-batch version, the approximation ratio depends on the batch size and initialization, which may not always guarantee optimal clustering.

Parameter Sensitivity: The performance heavily relies on parameters such as batch size and learning rate, which need careful tuning.

Complexity in Implementation: Implementing the recursive distance update and maintaining inner products can be intricate, potentially increasing the implementation complexity.

Potential Issues

Stochastic Nature: The inherent stochasticity of mini-batch algorithms can lead to variations in performance, and convergence to local minima is not guaranteed.

Parameter Initialization: Poor initialization of cluster centers can significantly affect the algorithm's performance and convergence speed.

Data Dependence: The effectiveness of the algorithm may vary depending on the dataset characteristics, such as the distribution and dimensionality of the data points.

**Questions:**

See Weaknesses above

**Limitations:**

Yes

---

> ### Author Rebuttal · Authors · 2024-08-05
>
> Thank you for your comments. Regarding your points:
>
> - The performance of the algorithm does not require careful tuning of the learning rate as you suggest since $\alpha_i^j$ is totally determined by the formula given at the end of page 6. Please can you clarify what you meant?
> - The approximation quality doesn't depend on the batch size as you suggest, we get it for free (in expectation) by initialising with kmeans++. Please can you clarify what you meant?
> - Please can you clarify what you mean by "convergence to local minima is not guaranteed"? We prove our algorithm stops due to early stopping after a constant number of iterations whp.
> - Poor initialization won't effect convergence speed due to the early stopping condition. Please can you clarify what you meant?

---

### Author Rebuttal · Authors · 2024-08-05

Following the reviewer comments, we ran additional experiments with graph datasets. Please find the details of the experiments below and a PDF with the results attached.


An advantage of kernel k-means compared to (non-kernel) k-means is its ability to handle graph datasets. Specifically, we can take a (potentiality sparse) graph as input and compute its Heat kernel (Spectral Graph Theory, Fan Chung, chapter 10) and run kernel k-means on top of that. We show experimentally that our algorithm is an order of magnitude faster than the full batch kernel k-means algorithm, with almost no loss in the quality of the clustering. The Heat kernel of a graph is defined as $H(t)\triangleq \exp(-t\mathcal{L})$ where $\mathcal{L}$ is the normalized Laplacian of the input graph and $t$ is a parameter.

Datasets: We create graph datasets by constructing $k$-nearest neighbour graphs for each dataset in our paper; then we compute the heat kernel for each. We use $t=8.0$ for all datasets, $k=1000$ for PenDigits, $k=500$ for letter and $k=250$ for HAR and MNIST. We do not count the time to construct the heat kernel in the runtimes of Figure 1 so the difference in runtime for different batch sizes will be more visible. Constructing the heat kernel took 1.2 seconds for PenDigits, 1.1 seconds for HAR, 19.6 for MNIST and 19.8 for Letter. We recorded the empirical values of gamma to be 0.036 for PenDigits, 0.060 for HAR, 0.055 for MNIST and 0.040 for Letter.

We believe this approach to be a more realistic test than the stochastic block model while also being small enough to run the full batch kernel k-means algorithm on. Any larger and it would simply take too long.

---

### Decision · Program_Chairs · 2024-09-25

**Decision:**

Reject

**Comment:**

While the reviewers generally recommended for acceptance, none were especially enthusiastic, and–as the authors acknowledged in their response–the reviewers had consistent concerns about theoretical novelty w.r.t. Schwartzman. In light of this concern, it might be a good idea to focus on polishing the experiments to further highlight the advantages of the approach.